# Translating staff experience into organisational improvement: the HEADS-UP stepped wedge, cluster controlled, non-randomised trial

Samuel Pannick,[1] Thanos Athanasiou,[2] Susannah J Long,[1,3] Iain Beveridge,[4] Nick Sevdalis[5]

► Prepublication history and additional material are available. To view these files please visit the journal online (http://dx.doi.org/10.1136/).

[1]NIHR Imperial Patient Safety Translational Research Centre, Imperial College London, London, UK
[2]Department of Surgery and Cancer, Imperial College London, London, UK
[3]Imperial College Healthcare NHS Trust, London, UK
[4]West Middlesex University Hospital NHS Trust, Isleworth, UK
[5]Centre for Implementation Science, Kings College London, London, UK

**Correspondence to**
Samuel Pannick; s.pannick@imperial.ac.uk

## ABSTRACT

**Objectives** Frontline insights into care delivery correlate with patients' clinical outcomes. These outcomes might be improved through near-real time identification and mitigation of staff concerns. We evaluated the effects of a prospective frontline surveillance system on patient and team outcomes.

**Design** Prospective, stepped wedge, non-randomised, cluster controlled trial; prespecified per protocol analysis for high-fidelity intervention delivery.

**Participants** Seven interdisciplinary medical ward teams from two hospitals in the UK.

**Intervention** Prospective clinical team surveillance (PCTS): structured daily interdisciplinary briefings to capture staff concerns, with organisational facilitation and feedback.

**Main measures** The primary outcome was excess length of stay (eLOS): an admission more than 24 hours above the local average for comparable patients. Secondary outcomes included safety and teamwork climates, and incident reporting. Mixed-effects models adjusted for time effects, age, comorbidity, palliation status and ward admissions. Safety and teamwork climates were measured with the Safety Attitudes Questionnaire. High-fidelity PCTS delivery comprised high engagement and high briefing frequency.

**Results** Implementation fidelity was variable, both in briefing frequency (median 80% working days/month, IQR 65%–90%) and engagement (median 70 issues/ward/month, IQR 34–113). 1714/6518 (26.3%) intervention admissions had eLOS versus 1279/4927 (26.0%) control admissions, an absolute risk increase of 0.3%. PCTS increased eLOS in the adjusted intention-to-treat model (OR 1.32, 95% CI 1.10 to 1.58, p=0.003). Conversely, high-fidelity PCTS reduced eLOS (OR 0.79, 95% CI 0.67 to 0.94, p=0.006). High-fidelity PCTS also increased total, high-yield and non-nurse incident reports (incidence rate ratios 1.28–1.79, all p<0.002). Sustained PCTS significantly improved safety and teamwork climates over time.

**Conclusions** This study highlighted the potential benefits and pitfalls of ward-level interdisciplinary interventions. While these interventions can improve care delivery in complex, fluid environments, the manner of their implementation is paramount. Suboptimal implementation may have an unexpectedly negative impact on performance.

## Strengths and limitations of this study

► This is the first controlled evaluation of a prospective frontline surveillance strategy on general medical wards.
► The pragmatic nature of the trial increases the generalisability of its findings.
► It can be difficult to disentangle the effects of interdisciplinary interventions from the characteristics of the teams that implement them best.
► This was a non-randomised study, adopting a necessarily pragmatic approach to the order in which participating teams introduced the intervention.
► Contamination between the groups, whereby PCTS generated organisational support for wards in the control period, may have reduced the intervention's measurable effect.

**Trial registration number** ISRCTN 34806867 (http://www.isrctn.com/ISRCTN34806867).

## INTRODUCTION

Frontline staff have a particular insight into the safety and quality of inpatient care. Favourable staff perceptions of care correlate with improved clinical outcomes, including patient survival.[1 2] Yet frontline concerns do not feature in national safety initiatives, nor are they a priority at a local level.[3 4] Senior leaders remain reluctant to pursue unvetted reports of frontline problems.[5] Better methods are needed to capitalise on frontline experiences and translate them into organisational action.

Prospective clinical surveillance systems can promote the systematic identification of safety concerns from a frontline perspective. Embedded observers, or visiting facilitators, work with frontline staff to form a structured record of their experience of care delivery— and its consequences. Observational studies

**Table 1** Institution characteristics

| | Hospital 1 | Hospital 2 |
|---|---|---|
| **Institution type (inpatient admissions/year)** | | |
| | Community hospital (52 000) | Academic teaching hospital (186 000) |
| **Participating wards (n)** | | |
| | Acute medical unit (2) | Geriatrics (1) |
| | Gastroenterology/internal medicine (1) | |
| | Heart failure/internal medicine (1) | |
| | Geriatrics (1) | |
| | Respiratory/internal medicine (1) | |
| **Existing interdisciplinary practice and approaches to adverse event detection and monitoring** | | |
| | Daily interdisciplinary 'board round,' typically focused on patients' discharge requirements | Daily interdisciplinary 'board round,' typically focused on patients' discharge requirements |
| | Online incident reporting system | Online incident reporting system |
| | Unstructured mortality case-note reviews | |
| **Major organisational changes during the study period** | | |
| | Incipient institutional merger | Deployment of a new electronic health record |

suggest this is a more engaging route to improvement than traditional incident reporting.[6 7] The appeal of these systems is clear. They may act at multiple organisational levels: (1) motivating and empowering staff to resolve unit-level issues within their control; (2) recording frontline successes and challenges to build a rich understanding of safety and resilience across the organisation; and (3) engaging leadership in attending to frontline concerns. Nonetheless, prospective clinical surveillance has not yet been tested as an intervention. Its impact on patient and team outcomes is unknown.

Here, we sought to evaluate prospective clinical *team* surveillance (PCTS), a novel extension of prospective clinical surveillance, on UK medical wards. PCTS combines structured, frontline interdisciplinary briefings with facilitated organisational escalation of the issues they identified, and feedback. PCTS was tested in a pragmatic, cluster controlled trial, assessing its effects on patient outcomes, and staff safety attitudes and behaviours.

## METHODS
### Study design and patients
We conducted a prospective, interventional, non-randomised stepped wedge trial on seven medical wards at two NHS (National Health Service) hospitals (table 1). The trial was described in a published protocol.[8] Interdisciplinary ward teams were assigned to a multifaceted quality improvement intervention. Wards introduced the intervention at staged intervals over the study period, such that (by the end of the trial) all teams had adopted the intervention and contributed both control and intervention group data. Stepped wedge designs are increasingly used to evaluate service-level interventions in acute care.[9–11] The stepwise implementation protocol is helpful when simultaneous rollout of the intervention would be

impractical for logistical reasons.[12] Baseline data collection began in August 2013, with implementation of the intervention between December 2013 and February 2015.

Medical ward teams with an existing structure for daily interdisciplinary team meetings, and their managers, were invited to take part. All the approached teams agreed to participate. The order in which wards adopted the intervention was pragmatically guided by local constraints. It would have been counterproductive to insist on a fully randomised implementation schedule. Instead, we sought input from senior ward staff and nursing leadership, trying to identify when they felt they could support the intervention's introduction. In practice, personnel and organisational changes meant that the anticipated leadership support could not be guaranteed. Similar pragmatic approaches have been used in other stepped wedge evaluations.[13] A schematic of the ward-level implementation schedule is provided in the online supplement (supplementary figure 1).

Adult patients admitted to participating wards during the study period were eligible. Exclusion criteria were: (1) less than 50% of the hospital admission spent on the specified ward; (2) discharge to a new skilled care facility (not the patient's previous address) or other hospital; (3) multiple ward transfers; (4) admission to the high dependency unit or intensive care unit (ICU); (5) elective admission, or direct admission from another hospital; and (6) surgeon-directed care for more than 24 hours of the admission. Other studies have used similar criteria to identify the patients whose outcomes might be most affected by service-level interventions.[14] The included patient group made up around 90% of all inpatients on these wards during the study period. Participating teams were teaching teams with attendings (consultants), residents (specialty trainees) and interns (foundation

doctors). Interdisciplinary staff also included ward clerks and nurses, who were typically based on a single ward. Doctors were based largely on one ward, but with some patient commitments elsewhere in the hospital. Physiotherapists and occupational therapists worked on multiple wards. Neither managers nor clinical staff had protected time for quality improvement initiatives.

## Intervention programme

### Structured team briefing

The PCTS programme was based on observational studies showing that medical teams can engage in the rapid identification and review of potential adverse events.[6 15–18] We designed a daily interdisciplinary briefing for structured team self-report on medical wards: the HEADS-UP (Hospital Event Analysis Describing Significant Unanticipated Problems) briefing. HEADS-UP briefings identified clinical and administrative challenges of the preceding 24 hours. They centred on the problems identified most commonly in the medical ward setting, as described in a recent, large, observational European study.[19] Structured team self-report captures the majority of errors documented by a trained observer.[20]

HEADS-UP briefings were co-developed with frontline clinical staff between October and November 2013. At the request of clinicians, a pro forma was developed with a visual format similar to the WHO's surgical safety checklist[21] (see online supplementary figure 2). Best practice recommendations for improvement interventions advise a degree of local flexibility[22]: other teams were able to make minor changes to their unit's pro forma, while retaining the overall format. This flexibility maintains the 'hard core' of the intervention, and permits a 'softer periphery' to maximise its uptake. Briefings ended with a team agreement on how to resolve, or escalate, the concerns that had been discussed. After a short period of supervision by the programme lead (a resident with a 0.75 full time equivalent commitment, working as an 'embedded researcher'[23] (SP)), the briefing was devolved to the ward team. Teams introduced daily HEADS-UP briefings between Monday and Friday, at the most appropriate time in their usual workflow. There was no protected time for team training. Briefings could be led by any member of the ward team, from ward clerk to therapist to attending physician.

### Facilitation

The second component of the intervention was facilitation, advancing the issues raised in the HEADS-UP briefings to bring about tangible unit-level and organisational changes. Facilitation helps to make sense of quality improvement interventions, aligning them with their participants and the surrounding context.[24] It is necessarily opportunistic and malleable, taking advantage of existing organisational levers. Here, it involved (for example) working with frontline teams to identify and document their areas of concern more effectively; championing those concerns in regular meetings with service leaders and safety committees; and following up on subsequent agreed actions when other priorities threatened their resolution. Other studies that have used a facilitator in a similar way have described the role as an 'animateur,' bringing on board people over whom he has no direct managerial authority.[25 26]

### Feedback

The third component of the intervention was feedback to participating teams, managers, governance committees and senior executives. Feedback summarised and disseminated the information collected in the daily briefings, highlighting HEADS-UP performance in each area, common concerns and challenges, and recurrent or unresolved problems that might require additional support. HEADS-UP data were provided on request to service leads, to support their business planning. System changes arising from HEADS-UP were publicised, for example, in existing departmental meetings, and via email and posters. As much as possible, feedback delivery was timely, focused on solution finding, signposted to relevant resources and adopted a non-judgemental approach.[27] Facilitation and feedback were provided by the programme lead.

Although described separately here, we hypothesised that the intervention components would be interlinked, in that changes arising from the programme's facilitation and feedback would motivate increasing engagement with HEADS-UP briefings, in turn increasing their ability to bring about change. We anticipated that improvements in interdisciplinary team care effectiveness would be brought about both by improvements in ward teams' function, and by incremental support service improvements in response to their concerns.

## Setting

Characteristics of the two participating institutions are provided in table 1. Most study wards were in a community general hospital in London. Both institutions faced significant challenges during the study period, with significant turnover in senior staff, mounting service pressures and financial restrictions. One hospital was in the process of a merger, and underwent a major inspection by the healthcare regulator (the Care Quality Commission). The other hospital was introducing an electronic health record.

## Study outcomes

The primary outcome was excess length of stay (eLOS). eLOS was a binary variable, representing an admission more than 24 hours above the patient's expected length of stay. Benchmarks for expected length of stay were generated using patient-level Healthcare Resource Groups[28] data from a network of four local hospitals. These hospitals were subject to the same community service restrictions and healthcare economy demands as the study sites. The study design therefore evaluated the extent to which intervention and control wards met this local standard.

Secondary outcomes included patient-level measures of (1) readmission within 30 days and (2) in-hospital death or death/readmission within 30 days. Ward-aggregate measures included (3) escalation events (referrals to the ICU outreach service, emergency calls and ICU transfers); (4) complications of care (pressure ulcers, *Clostridium difficile* infections, and methicillin-resistant *S. aureus* [MRSA], methicillin-sensitive *S. aureus* [MSSA] and *Escherichia coli* bacteraemias); and (5) incident reporting characteristics. Safety and teamwork climates were assessed at baseline and 6 months into the intervention period, using the relevant subscales from the Safety Attitudes Questionnaire.[29 30] Secondary outcomes (and the per protocol analysis described below) were prespecified.

### Data collection
Anonymised patient-level and ward-level outcomes were extracted from routinely collected data sets. Potential participants in the HEADS-UP briefings were invited to complete anonymous surveys at baseline and 6 months later, either by submitting responses electronically (via Survey Monkey) or via a paper questionnaire. Overall response rates were calculated using contemporaneous staffing rosters.

### Sample size
Power calculations were conducted for the primary outcome (eLOS) using a recommended methodology for stepped wedge trials.[31] As described in detail in the published protocol,[8] we adopted an intraclass correlation coefficient (ICC) of 0.06, based on the ICCs for length of stay and appropriateness of stay in trials of acute care pathways.[32] We iterated calculations based on the wards with the highest and lowest baseline eLOS rates.[7] With a two-sided p<0.05, a sample size of 7840 patients was needed to detect a 2%–14% absolute risk reduction, with a power between 75% and 100%.[8] This approach to sample size calculations and study power has been used for other stepped wedge trials.[33]

### Allocation and blinding
Individual patients were not recruited separately to the study, so we did not anticipate significant bias due to lack of allocation concealment.[34] Staff could not be blinded to their ward's assignment, due to the nature of the intervention. Clinical outcome data and escalation of care data were extracted by local administrative staff blinded to the study, as part of their ordinary duties. Hospital peer group data were generated by the data extraction services *CHKS* (Alcester, UK) and *Dr Foster* (London, UK), also blinded to intervention group.

### Statistical analysis
Multilevel mixed-effects models (Stata V.14.2) were used to evaluate the intervention's effect on each patient-level and ward-level outcome. This statistical approach accounts for clustering of outcomes within wards, and repeated measurements over time, representing ward-level variance in patient outcomes as a random effect. For the binary patient-level outcomes, binary logistic models were used; for counts data (complications of care and processes of care), we used Poisson loglinear models. Analyses involving eLOS were restricted to those patients who survived to discharge; no patients were excluded because of 'outlier' length of stay. General linear regression models were used to analyse survey data using a difference-in-differences approach[35] with an interaction code for time*intervention at each site (SPSS, V.22). This method evaluated whether changes in survey responses over time differed between PCTS participants and non-participants.

In addition to the intention-to-treat analysis, we conducted a per protocol analysis, evaluating the effect of briefing implementation fidelity. Based on observations during the pilot period, we described implementation fidelity each month as the product of briefing frequency and engagement. High fidelity required both high briefing frequency and correspondingly intensive documentation by the ward. High frequency was coded if briefings took place on ≥75% working days that month. Monthly engagement was defined as high if teams documented more than the median number of issues. An interaction term 'frequency*engagement' was incorporated into each model, with a code for high fidelity where both engagement and frequency were high. Note that this per protocol analysis, which calculated different implementation fidelity codes for each ward month, allowed for varying fidelity by a single ward team over the course of the study.

Patient-level outcomes were adjusted for time effects, age, Charlson comorbidity index,[36] and palliation status, as well as ward admissions. Time effects were specified as a continuous variable coded for each study month. Ward-aggregate outcomes were adjusted for time effects, seasonal trends, the median Charlson score of patients on the ward that month and the rotation of interns between departments. Survey analysis defined HEADS-UP participation as self-reported engagement in five or more briefings, and adjusted for self-reported workload on the NASA Task Load Index scale,[37] as well as hospital site and time period.

### Ethics and consent
The Research and Development authority at each participating institution approved the study as a service development initiative, exempting it from formal ethical evaluation. As noted in the study protocol, the Office for Regulatory Compliance at the Imperial College Academic Health Science Centre initially advised that registration with a clinical trial database was not required for a study of this nature.[8] This decision was reviewed at the authors' request, and the study was registered—prior to completion of data collection—in the ISRCTN registry (https://dx.doi.org/10.1186/ISRCTN34806867).[8] Staff were aware that the service development was being formally evaluated. As is routine for this type of intervention, we did not seek participant-level consent.

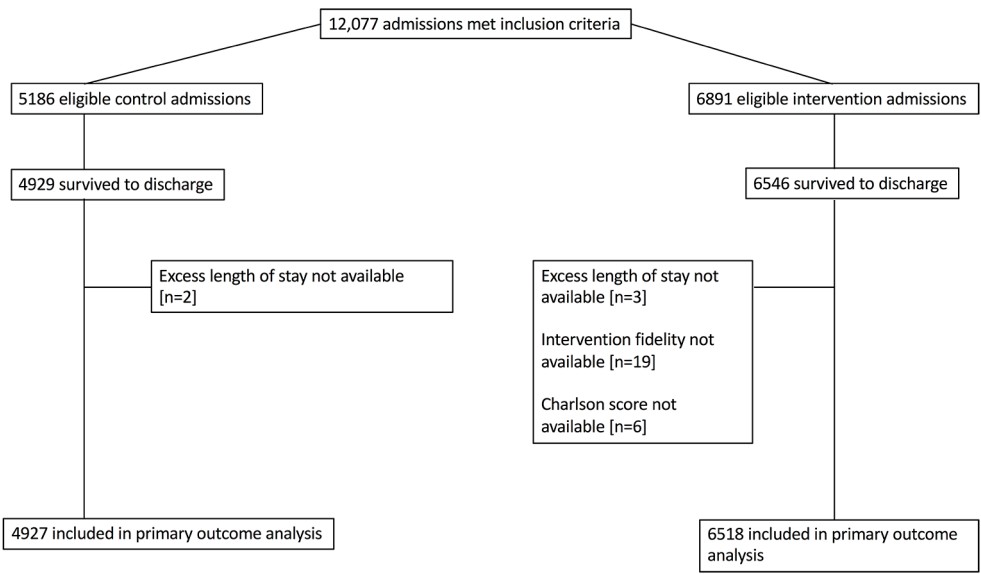

**Figure 1** Patient flow diagram.

## RESULTS

A total of 12 077 eligible admissions took place during the study period, of which 11 445 were included in the primary analysis (figure 1). A total of 4927 admissions were in the control group and 6518 admissions were in the intervention group. Monthly ward admissions were higher in the intervention months (77 admissions/ward/month vs 64 admissions/ward/month), as was unadjusted length of stay (2.25 days vs 2.0 days) (table 2).

### Implementation fidelity

Briefing implementation data were available for 71/73 (97.2%) ward months. There was variation in implementation fidelity, both in terms of briefing frequency (median 80% working days/month, IQR 65%–90%)

**Table 2** Ward and admission characteristics during the intervention period. Values are median (IQR) unless otherwise stated.

| | Control group | Intervention group |
|---|---|---|
| **Ward characteristics** | | |
| Number of patient admissions | 4927 | 6518 |
| Ward admissions per month | 64 (55–153) | 77 (64–133) |
| Length of stay for study patients (days) | 2.0 (0.8–7.6) | 2.2 (0.9–7.3) |
| **Admission characteristics** | | |
| Age (years) | 67 (49–80) | 65 (47–78) |
| Age ≥65 years (n, %) | 2707 (54.9) | 3275 (50.2) |
| Female (n, %) | 2537 (51.5) | 3391 (52.0) |
| Charlson comorbidity index | 3 (0–10) | 3 (0–8) |
| Palliative coding (n, %) | 50 (1.0) | 94 (1.4) |

and engagement (median 70 issues/ward/month, IQR 34–113). In the intervention group, 3607/6518 (55.3%) admissions were in high-fidelity ward months.

Successful implementation at hospital 1 was recognised by the Care Quality Commission (the UK's healthcare quality inspectorate), whose inspectors observed briefings in use and formally reported them as 'outstanding practice,' their highest rating. More limited organisational attention at site 2 resulted in slower programme implementation. At that site, PCTS was only embedding into routine practice as the second survey period began.

### Primary outcome: eLOS

A total of 1714/6518 (26.3%) intervention admissions had eLOS versus 1279/4927 (26.0%) control admissions, an absolute risk increase of 0.3% (table 3). In the unadjusted model, eLOS did not change significantly with PCTS (OR 1.07, 95% CI 0.96 to 1.19, p=0.219). When adjusted for patient-level and ward-level covariates, the intention-to-treat model showed increasing eLOS with PCTS (OR 1.32, 95% CI 1.10 to 1.58, p=0.003). Of note, there was an underlying trend of improved performance as the study period progressed (OR 0.98, 95% CI 0.96 to 0.996, p=0.016). The ICC for eLOS was 0.07 (95% CI 0.018 to 0.250).

The apparent worsening of performance with PCTS did not reflect whether, or how, the intervention had been used. In the planned per protocol analysis, high-fidelity PCTS implementation significantly reduced eLOS: OR 0.79, 95% CI 0.67 to 0.94, p=0.006. A sensitivity analysis (coding time effects as a categorical variable) confirmed the effect sizes of both intention-to-treat and per protocol analyses, and their statistical significance.

An exploratory plot of the modelled relationship between engagement, briefing frequency and the probability of eLOS highlighted different outcomes with high-fidelity implementation and lower fidelity

**Table 3** Summary of analysis for primary and secondary outcomes

| Outcome | PCTS | Control | Adjusted outcome ratio* | | | | |
| --- | --- | --- | --- | --- | --- | --- | --- |
| | | | Intention-to-treat | | High fidelity† | | |
| | | | Outcome ratio (95% CI) | p Value | Outcome ratio (95% CI) | p Value | |
| **Primary outcome** | | | | | | | |
| Excess length of stay, n (%) | 1714 (26.3) | 1279 (26.0) | 1.32 (1.10 to 1.58) | 0.003 | 0.79 (0.67 to 0.94) | 0.006 | |
| **Secondary outcomes** | | | | | | | |
| Readmission within 30 days, n (%) | 950 (14.6) | 565 (11.5) | 1.12 (0.91 to 1.37) | 0.296 | 1.11 (0.93 to 1.32) | 0.247 | |
| In-hospital death or death/readmission within 30 days, n (%) | 1657 (24.2) | 1190 (23.0) | 1.03 (0.86 to 1.22) | 0.759 | 1.07 (0.92 to 1.25) | 0.363 | |
| Escalation events/month, median (IQR)‡ | 8 (3.5–16) | 3 (1–6) | 0.95 (0.74 to 1.23) | 0.713 | 1.01 (0.83 to 1.24) | 0.919 | |
| Complications of care/month, median (IQR) | 1 (0–2) | 1 (0–3) | 0.92 (0.56 to 1.50) | 0.727 | 1.28 (0.76 to 2.13) | 0.351 | |
| Total incident reports/month, median (IQR) | 16 (10–23.5) | 15.5 (9.25–20) | 0.97 (0.82 to 1.14) | 0.673 | 1.28 (1.12 to 1.47) | <0.001 | |
| High-yield incident reports/month, median (IQR)§ | 11 (6–17.5) | 9 (5–14.75) | 0.92 (0.75 to 1.12) | 0.400 | 1.40 (1.19 to 1.65) | <0.001 | |
| Non-nurse incident reports/month, median (IQR) | 2 (0.5–4) | 1 (1–2.75) | 0.89 (0.55 to 1.42) | 0.618 | 1.79 (1.24 to 2.58) | 0.002 | |

Patient-level outcomes were adjusted for time effects, age, Charlson comorbidity index, and palliation status, as well as ward admissions. Ward-aggregate outcomes were adjusted for time effects, seasonal trends, the median Charlson score of patients on the ward that month and the rotation of interns between departments.

*OR for binary outcomes; incidence rate ratio for continuous outcomes.

†High fidelity = high frequency (≥75% working days) and monthly reporting ≥ median issues.

‡Escalation events = emergency calls, intensive care unit transfers and referrals to the intensive care unit outreach service.

§High-yield incident reports = not relating to slips, trips or falls.

PCTS, Prospective clinical team surveillance

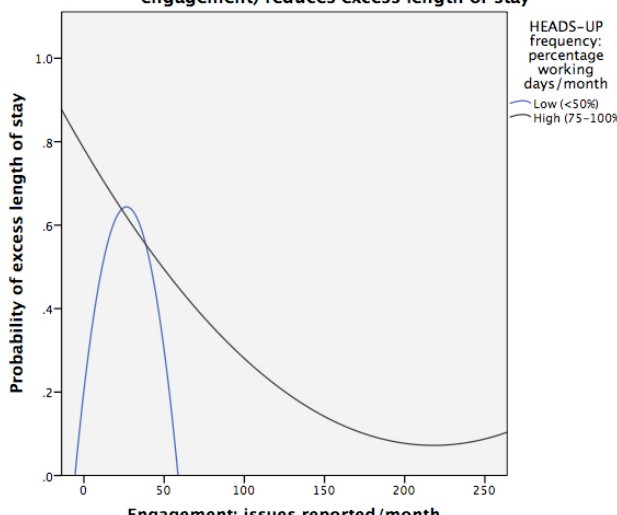

**Figure 2** Relationship between engagement and frequency of HEADS-UP briefings, and the probability of excess length of stay. HEADS-UP, Hospital Event Analysis Describing Significant Unanticipated Problems.

implementation (figure 2). This plot models engagement and briefing frequency as continuous variables within a multilevel model to help visualise their interaction.

### Secondary outcomes

No changes were seen in readmissions or the composite of deaths/readmissions, irrespective of implementation fidelity (all CIs include unity; all p values>0.2). Escalation events and complications of care were also unchanged (table 2).

There was no change in incident reporting in the intention-to-treat analysis. By contrast, high-fidelity implementation increased the total number of incident reports submitted each month (incidence rate ratio [IRR] 1.28, 95% CI 1.12 to 1.47, p<0.001), with increases in non-falls incident reports (IRR 1.40, 95% CI 1.19 to 1.65, p<0.001) and reports from non-nursing staff (IRR 1.79, 95% CI 1.24 to 2.58, p=0.002).

Survey response rates were 71.8% (61/85) at baseline and 65.1% (54/83) at 6 months, with 97.7% safety and teamwork questions completed. The respondent demographics are provided in the online supplementary table 1. Unadjusted mean safety and teamwork scores were non-significantly higher after PCTS participation (table 4). With sustained implementation, PCTS improved adjusted safety and teamwork scores over time (site 1 estimated marginal mean (EMM) safety score 70.3 vs 61.0, p=0.021; EMM teamwork score 81.7 vs 70.0, p=0.004). The significant improvement persisted in a sensitivity analysis that adjusted survey results for ward-level as well as hospital-level clustering. 71.9% (23/32) of site 1 respondents agreed they had reported concerns differently as a result of the programme. Of note, higher workload scores correlated with lower safety scores (B=−1.52, 95% CI −2.59 to −0.44, p=0.006) but not teamwork scores (B=−0.21, 95% CI −1.10 to 0.68, p=0.64).

### DISCUSSION

In this stepped wedge cluster controlled trial, intention-to-treat and per protocol analyses produced conflicting results. eLOS increased overall with PCTS, but was reduced by high-fidelity PCTS implementation. Among

| Table 4 | Effect of PCTS on safety and teamwork attitudes | | | |
|---|---|---|---|---|
| | **PCTS** | | | |
| | | **Participants** | **Non-participants** | **p Value** |
| Mean safety score (SD) | | | | |
| | | 68.4 (13.6) | 65.9 (16.9) | 0.667 |
| Mean teamwork score (SD) | | | | |
| | | 80.2 (12.1) | 79.6 (12.3) | 0.524 |
| Estimated marginal mean safety score (95% CI) | | | | |
| Hospital 1 | Baseline | 59 .8 (51.4 to 68.1) | 69.2 (64.6 to 73.9) | 0.021* |
| | 6 months | 70.3 (65.3 to75.2) | 61.0 (50.8 to 71.2) | |
| Hospital 2 | Baseline | 70.8 (51.1 to 90.4) | 54.9 (46.2 to 63.7) | |
| | 6 months | 71.0 (56.0 to 86.0) | 75.1 (63.5 to 86.7) | |
| Estimated marginal mean teamwork score (95% CI) | | | | |
| Hospital 1 | Baseline | 72.3 (65.3 to 79.2) | 82.6 (78.8 to 86.5) | 0.004* |
| | 6 months | 81.7 (77.6 to 85.8) | 70.0 (61.6 to 78.4) | |
| Hospital 2 | Baseline | 84.8 (68.5 to 100.0†) | 74.5 (67.2 to 81.7) | |
| | 6 months | 88.2 (75.7 to 100.0†) | 83.3 (73.7 to 92.9) | |

*p value for model effect of interaction term PCTS participation*hospital*time
†The upper bound of the 95% CI for estimated marginal mean climate scores was truncated at 100, the limit of the scale.[49]
PCTS, prospective clinical team surveillance.

the secondary outcomes, safety and teamwork climates improved with sustained PCTS implementation. High-fidelity PCTS increased both incident reporting by non-nursing staff, and the number of non-falls reports.

There are several possible explanations for the tension between the two analyses. First, faithful attention to quality improvement efforts may indeed result in worse outcomes, perhaps by distracting attention from existing good practice.[38] This is unlikely to have been the mechanism here: the wards that dedicated most time and effort to the intervention saw improved outcomes. Second, there may have been unmeasured confounders distinguishing between high-fidelity and low-fidelity PCTS teams. Teams that participate more wholeheartedly in trials may be more innovative, with stronger leadership, a readiness for change and better managerial relations.[39] It is difficult, therefore, to entirely separate the intervention's effects from the characteristics of the teams that implemented it best. However, the study design mitigated for this, allowing for varying implementation fidelity by a single team across the study period. As the design elicited high-fidelity ward *months* rather than high-fidelity teams per se, unmeasured team characteristics should hold less influence over the results. It is unlikely that our findings were a simple reflection of teams' pre-existing practice or fluctuations in their workload.

The most likely explanation is that the intervention was used differently by high-fidelity and low-fidelity wards. In the linked qualitative analysis, high-fidelity PCTS improved team autonomy in resolving problems, and facilitated managerial resolution where necessary.[40] By contrast, low-fidelity teams may have used PCTS simply to record issues with a degree of frustration, without any positive change in their attitudes or behaviours.[40] Rather than enacting change for themselves, low-fidelity teams perhaps deferred to facilitation and managerial input, believing that they would be sufficient to address unit-level problems. Previous quality improvement programmes have also inadvertently blurred lines of responsibility.[41] While imperfect PCTS implementation might be deleterious, our results suggest that interdisciplinary quality and safety interventions—when implemented well—remain a viable route to improvement.

Earlier studies of prospective clinical surveillance did not evaluate an effect on patient outcomes, and so we can only position our work within a broader literature on ward-level interdisciplinary interventions. The variable implementation of these interventions likely contributes to their disputed impact on length of stay.[42 43] Overall, the experience of PCTS implementation mirrored that of other structured team initiatives. Only 62% of mandated interdisciplinary surgical checklists are completed in their entirety.[44] Similarly, structured bedside rounds may be implemented for only 54% of patients, despite co-creation with frontline staff.[45] The benefits of these interventions are only identified at high fidelity,[44] and so optimising their implementation should remain a focus for further research.

This study's strengths include its pragmatism: the trial had broad eligibility criteria; a typical clinical setting; little additional investment for study recruitment or follow-up; and flexibility with regard to intervention delivery and adherence.[46] Limitations include the difficult balancing act between trial rigour and implementation effectiveness. Attempts to preserve distinct intervention and control groups may have constrained implementation fidelity, limiting a more natural, collaborative spread of practice. Conversely, contamination between the groups may have reduced the intervention's measurable effect: PCTS-generated organisational support (over which we had no control) was not necessarily directed to areas that first raised concerns. Last, there may have been inadequate adjustment even for known confounders like workload. Ward admissions are an imperfect proxy for workload: demands on staff increase as the numbers of admissions, transfers and discharges all increase.[47] Ward-level staffing data are rarely available in this detail in the UK.[48]

## CONCLUSIONS

Overall, PCTS increased eLOS for general medical patients - a worsening of the primary outcome. In contrast, high-fidelity PCTS reduced eLOS, improved safety and teamwork attitudes, and increased high-yield incident reporting. Our results suggest that interdisciplinary interventions can improve care delivery in complex, fluid environments like medical wards. The manner in which these interventions are implemented— honouring the spirit of the interventions and replicating their proposed mechanism of actions—is vitally important. Suboptimal implementation may have an unexpectedly negative impact on team performance.

**Contributors** Study design: SP, TA, NS. Study implementation: SP, SJL, IB. Statistical analysis: SP, TA. All authors contributed to, read and approved the final manuscript. SP had full access to all of the data in the study and takes responsibility for the integrity of the data and the accuracy of the data analysis.

**Funding** This paper represents independent research supported by the National Institute for Health Research (NIHR) Imperial Patient Safety Translational Research Centre, Imperial College Healthcare Charity (Grant GG14\1022) and West Middlesex University Hospital NHS Trust. NS' research is supported by the NIHR Collaboration for Leadership in Applied Health Research and Care South London at King's College Hospital NHS Foundation Trust. NS is a member of King's Improvement Science, which is part of the NIHR CLAHRC South London and comprises a specialist team of improvement scientists and senior researchers based at King's College London. Its work is funded by King's Health Partners (Guy's and St Thomas' NHS Foundation Trust, King's College Hospital NHS Foundation Trust, King's College London and South London and Maudsley NHS Foundation Trust), Guy's and St Thomas' Charity, the Maudsley Charity and the Health Foundation. No funding source had any role in the design and conduct of the study; collection, management, analysis or interpretation of the data; or preparation, review or approval of the manuscript. The views expressed are those of the authors and not necessarily those of the NHS, the NIHR or the Department of Health.

**Competing interests** NS is the director of London Training &Safety Solutions Ltd, which delivers team assessment and training to hospitalson a consultancy basis.

**Provenance and peer review** Not commissioned; externally peer reviewed.

**Data sharing statement** Anonymised data can be requested from the corresponding author.

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
