## [Reviewer comments · BMJ Open]

ARTICLE DETAILS

TITLE (PROVISIONAL)	Translating staff experience into organisational improvement: the HEADS-UP stepped wedge, cluster controlled, non-randomised trial
AUTHORS	Pannick, Samuel; Athanasiou, Thanos; Long, Susannah; Beveridge, Iain; Sevdalis, Nick

VERSION 1 - REVIEW

REVIEWER	Judith Poldervaart Patient safety/General practice, Julius Center, UMC Utrecht, the Netherlands
REVIEW RETURNED	12-Oct-2016

GENERAL COMMENTS	- Abstract: ranges of working days/month and issues/ward/month are fairly large, suggesting large differences in adherence of the briefing and participation of the staff. Something you hopefully address in results/discussion- Abstract: PCTS had no impact, however, only in adjusted per protocol analysis, HIGH FIDELITY PCTS reduced eLOS. In my opinion, this conclusion is somewhat misleading. I suggest rewriting the conclusion of the abstract.- Introduction: "The effects of interdisciplinary team interventions in general medical settings have been mixed,12,13 and recent editorials have called for adequately powered studies with designs that better account for confounding factors.14"  the reader now cannot judge for themselves what these mixed results mean or how big the differences are. Include more detail. The same for the confounding factors; mention one or two factor, to help the reader understand the problem.- introduction: Research question is a bit hidden, with the sentence beginning with "here, we....". Also it misses a link with the lines before on the problems with the previous research. Please rewrite: We aimed to investigate.... (or something similar), indicating the research aim. Also, provide a linking sentence, addressing the link between the limitations of previous research and the current aim.- Methods: Study design and patients is not the first paragraph. I miss this information while reading the intervention. Please reverse. I see no reason why this is done, there is no information on intervention needed to understand the paragraph "Study design and patients"- Methods: pag 8 "other teams were able to tailor their unit's pro forma whilst retaining the overall format"  how can this have influenced the results? I can imagine with more tailoring, the effect can differ per setting, making it less generalizable in other settings?- Methods: pag 9: as much as possible, feedback was delivered timely.  the reader cannot know what is meant by timely. Please address what range existed during the trial, since this will make it
--

	possible for other researchers to implement the intervention according to the standards you used.  - Methods: Exclusion criteria: for me it is not clear why these exclusion criteria were applied, and this will reduce applicability in real life, since there a ward contains a lot of different types of patients. Please explain. - Methods: Ethics: there is no ethics subheading, please provide. Furthermore, "The study was approved as a quality improvement program by research and development authorities at participating institutions"  it is unclear why there has been no ethical committee? Please also provide reasons for this. There is no institution who reviewed the protocol? Please provide names of institution for that. Has the trial been registered at Clinicaltrials.gov or any other registration? Please comment. - methods: outcomes: Local institutions contributing comparator data....  this confuses me, since it is a controlled trial already. So you also wished to adjust for baseline with a comparable other setting, who did not receive the intervention at all? Please explain, in rebuttal and in manuscript. - Methods: sample size: "We adopted a conservative intraclass correlation coefficient of 0.06"  how did you arrive at this ICC? please provide additional information in manuscript. - methods: sample size: "baseline outcome rates" -- > what is meant by this? I dont know what is this referring to? Primary outcome of eLOS? - Methods: sample size: 7840 eligible patients  so this is the sample size? It is not clear from your text if you mean: if they do that, OR if you mean: we have wards that include 7840 patients? - Methods: allocation: Hospital peer group data were generated by CHKS (Alcester, UK) and Dr Foster (London, UK..."  I have no idea what these CHKS and dr foster are? Dataextraction services? Is dr foster a person? Please provide more info. - Results: pag 12 "At that site, PCTS was only embedding..."  embedded - Results: pag 13: "in the adjusted intention-to-treat model, the intervention was associated with increasing eLOS."  what adjustment was made? It stands in the methods i think, but know readers do not see what is adjusted for. - Discussion: "In a systematic review, 70% of interdisciplinary interventions had no effect on the length of stay;12 those that did were typically high intensity, facilitated interventions like PCTS."  if there is a review on this topic with several trials, what exactly does this current research add? - References: A number of 57 references is too much. Please adjust to 40 as a maximum.
--	--

REVIEWER	Robert C Amland Cerner Corporation, USA
REVIEW RETURNED	01-Nov-2016

GENERAL COMMENTS	This study is impressive throughout; classic stepped wedge trial study design.
--

REVIEWER	Karla Hemming University of Birmingham
-----------------	---

	UK
REVIEW RETURNED	15-Nov-2016

GENERAL COMMENTS	This is a clear write up of a nice study. The authors have overlooked reporting a few important items (see below) and I hope that by adding these extra pieces of information will help the authors complete a full description of their trial. I think there is one very important oversight - the authors have failed to adjust for time in their analysis and this should be the primary treatment effect estimate. The study is also non-randomised - this should be clearer and some discussion of the limitation of this is needed. Translating staff experience into organisational improvement: the HEADS-UP stepped wedge cluster controlled trial Review by Karla Hemming November 2016  1. Title needs clarity on whether the study is randomised. Turns out not to be randomised, I think the authors should be explicit about this in the title. Likewise abstract. 2. Abstract needs to include information on the number of transition steps. Also crucial to report the time adjusted treatment effect in the abstract, and clearly report that this is the primary treatment effect from the trial. (Others will be possibly biased). The abstract doesn't include any eligibility criteria for the clusters (neither does paper) I suspect this is because this is an evaluation of a routine roll-out or something like that, but this should be clear. 3. The summary strengths and limitations section doesn't mention that this is a non-randomised evaluation. It also does not follow why the trial is "well powered". The CIs are actually quite wide – large cluster sizes don't necessarily translate into lots of power. I suggest this statement is removed. 4. In the main paper the background should include justification for use of a cluster design and use of a stepped wedge design. 5. The objectives need to be clearer and more specific. 6. A description of setting / control condition is missing. 7. A schematic representation of the study is needed. How long were the steps? How many clusters transitioned at each step? What was the order of the transitions? (Some of this information may be too tricky to write in text, but could be nicely summarised in a picture). 8. More explicit detail is needed on how this is a non-randomised study, and information needed on how the order was determined. 9. Page 9 – line 17, 18 – is this equivalent to research ethics approval? Who was consented? Were health care professionals told they were part of an evaluation and was there consent taken? How about patients? Was the data all anonymous and routinely collected? 10. Page 9 line 28 to 32 – unclear what this comparator data
---

	is???  11. Why an ICC of 0.06? Reference 34 doesn't clearly link to a sample size methodology. Need to report baseline (control) proportions. Power calculation cannot currently be replicated – need detail on design. (See above) 12. Statistical methods – need to adjust for time effects – this is crucial and is the most important limitation of the paper. 13. ICCs need to be reported in the results – at least for the primary outcome. 14. Need absolute measures of treatment effects. 15. Was the trial registered (sorry if I missed this) 16. Can you clarify that all sub-groups and secondary outcomes were prespecified
--	---

REVIEWER	Jennifer A Thompson London School of Hygiene and Tropical Medicine, UK
REVIEW RETURNED	15-Nov-2016

GENERAL COMMENTS	This is an interesting article reporting the results of a non-randomised SWT. There are some major changes necessary in how the trial is reported in order for it to be a fair representation of the trial results. Please note that I have focused my review on statistical aspects of the article. I have not commented on the specific subject area of the trial. Major comments: It needs to be made clearer throughout the paper that the hospital wards were not randomly allocated. This is alluded to but is never explicitly stated. Please add to the trial limitations in the discussion that the trial was not randomised. The conclusions of the paper do not focus on the primary outcome, instead they focus on the per-protocol results. The main conclusion of the paper should be that overall, you found evidence that PCTS increased eLOS. The discussion can go on to explore the reasons for this. The abstract doesn't give the primary outcome. Your per-protocol analysis is written more like a subgroup analysis. In the protocol paper you define per-protocol to be high fidelity, but in table 3 you also present moderate fidelity results (but not low fidelity) and you give a p value for how fidelity affected eLOS. Please clarify if this is a per-protocol analysis, in which case remove moderate fidelity, or a subgroup analysis, in which case add interaction p values and low fidelity should be reported. I am concerned that results have been reported very selectively based on the direction of effect and p-values. For example, unadjusted analyses are presented for the intention to treat analysis rather than the pre-specified adjusted analysis, in the abstract the adjusted analysis result is not given at all. Minor comments:
--

	Please mention in the introduction that the context for this trial is NHS hospital wards, this isn't mentioned until the methods. Please add some more detail to the study design section. Although, you do state that there were 7 wards, you do not state how many steps the SWT had. How many months were there between each ward getting the intervention? How much follow up was there before any wards had the intervention and how long once all wards had the intervention? The reference for the sample size calculation seems odd; this is a review paper so it isn't clear what method you used. What do you mean by the sentence "Other stepped wedge trials have similarly produced a range within which the study power might lie"? Please add more details for how the survey data were collected and analysed. It looks like you had about 10 per ward? Were baseline results treated as covariates in the analysis or did you do a difference of differences analysis? How did you justify ignoring clustering? How was the absolute percentage reduction (page 12 line 31) calculated? This should be described in the statistical methods. Page 11 line 45, "data were available for 71/73..." why were there 73 ward months? It is not clear how long each ward spent in control and then intervention, so it is not clear how you end up with 73 ward months. What is the 80.7 issues/ward/month? Is that a mean? I'm also not sure how informative the range is, perhaps an IQR would give a better feel for the spread of the data? I couldn't see a reported ICC in the results, please add this. It would be useful to have a footnote to table 3 giving what the analyses adjusted for so the reader doesn't have to refer back to the methods. I found figure 1 difficult to understand, these look like modelled results rather than a plot of the data but there are no details about how this was done. Why isn't there a moderate line? Why does the low line increase and then decrease? In the abstract, what do you mean by "mixed methods evaluation"? There was no qualitative component addressed in this article.
--	---

VERSION 1 – AUTHOR RESPONSE

Reviewer 1:

1. Ranges of fidelity are fairly large, suggesting large differences in adherence of the briefing and participation of the staff – something to be addressed in results/discussion.

The updated manuscript explores the nature of intervention fidelity, and its impact, in more detail.

The most likely explanation is that the intervention was used differently by high fidelity and low fidelity wards. In the linked qualitative analysis, high fidelity PCTS improved team autonomy in resolving problems, and facilitated managerial resolution where necessary.² By contrast, low fidelity teams may have used PCTS simply to record issues with a degree of frustration, without any positive change in their attitudes or behaviours.² The differential uptake of improvement interventions by teams that are already performing well is a difficult challenge to resolve.³ Rather than enacting change for themselves, low fidelity teams perhaps deferred to facilitation and managerial input, believing they would be sufficient to address unit-level problems. Previous quality improvement programmes have also inadvertently blurred lines of responsibility.⁴²

2. Only high fidelity PCTS reduced eLOS – the Abstract conclusion should be rewritten.

As the reviewer suggests, we have revised the Abstract for a more accurate representation of the intention-to-treat results.

This study highlighted the potential benefits and pitfalls of ward-level interdisciplinary interventions. Whilst these interventions can improve care delivery in complex, fluid environments, thereby benefiting team and patient outcomes, the manner of their implementation is paramount. Suboptimal implementation may have an unexpectedly negative impact on performance.

3. Study design should be at the beginning of the Methods section.

We have altered the order of the Methods section to incorporate this change.

4. *Other teams were able to tailor their unit's pro forma whilst retaining the overall format. How can this have influenced the results or their generalisability?*

Each team's changes to their pro forma were minor, as we explain in the manuscript, and did not result in any significant deviation from the core components of the team briefing. The ability to make these minor changes was an important feature of the intervention, as teams need to feel ownership of these interventions with the right to iterate changes. This is a recommended feature of this type of interdisciplinary intervention; rather than making it less generalisable, it is a feature that future efforts would need to replicate.

Best practice recommendations for improvement interventions advise a degree of local flexibility:²³ other teams were able to make minor changes to their unit's pro forma, whilst retaining the overall format. This flexibility maintains the 'hard core' of the intervention, and permits a flexible 'soft periphery' to maximise its uptake.

5. *As much as possible, feedback was timely. The reader cannot know what is meant by timely. Please address what range existed during the trial.*

We describe the flexible and optimistic nature of feedback during the trial, delivered continuously over 15 months to participating teams, managers, committees and executives. The feedback was delivered in person, by email and posters. Clearly, this cannot be summarised in any quantitative fashion. Rather, we define the principles of facilitation and feedback, which we would expect replication efforts to address.

Facilitation is necessarily opportunistic and malleable, taking advantage of existing organisational levers. Here, it involved (for example) working with frontline teams to identify and document their areas of concern more effectively; championing those concerns in regular meetings with service leaders and safety committees; and following up on subsequent agreed actions when other priorities threatened their resolution. Other studies that have used a facilitator in a similar way have described the role as an '*animateur*', bringing on board people over whom he has no direct managerial authority.^{5,6}

The third component of the intervention was feedback to participating teams, managers, governance committees, and senior executives. Feedback summarised and disseminated the information collected in the daily briefings, highlighting HEADS-UP performance in each area, common concerns and challenges, and recurrent or unresolved problems that might require additional support. HEADS-UP data were provided on request to service leads, to support their business planning. System changes arising from HEADS-UP were publicised, e.g., in existing departmental meetings, and via email and posters. As much as possible, feedback delivery was timely, focused on solution-finding, signposted to relevant resources, and adopted a non-judgmental approach.⁷ Facilitation and feedback were provided by the programme lead.

6. *Why were these exclusion criteria applied, and will this reduce applicability in real life?*

We now reassure the reader that these exclusion criteria are appropriate for this type of intervention, whilst still allowing the study to include the vast majority of ward patients.

Other studies have used similar criteria to identify the patients whose outcomes might be most affected by service-level interventions.⁸ The included patient group made up around 90% of all inpatients on these wards during the study period.

7. *There is no ethics subheading; please explain why there was no ethical committee and provide the name of the institution who reviewed the protocol, and address the trial's registration.*

As we noted in the trial protocol, the intervention was approved (by two local Research and Development authorities) as a service development initiative not requiring a formal ethical assessment. This follows a common model through which similar interventions have been approved. The trial was formally registered, prior to completion of data collection, with a delay that was outside the authors' control. We replicate this information from the protocol so that the ethics and registration of the study are more easily accessed in this stand-alone paper. The trial registration was provided in the original manuscript submission (although it may have been visible only to the Editors rather than the reviewers) and we provide it again here.

The Research and Development authority at each participating institution approved the study as a service development initiative, exempting it from formal ethical evaluation. As noted in the study protocol, the Office for Regulatory Compliance at the Imperial College Academic Health Science Centre initially advised that registration with a clinical trial database was not required for a study of this nature.¹ This decision was reviewed at the authors' request, and the study was registered – prior to completion of data collection – in the ISRCTN registry (<https://dx.doi.org/10.1186/ISRCTN34806867>).¹ Staff were aware that the service development was being formally evaluated. As is routine for this type of intervention, we did not seek participant-level consent (from patients or staff).

8. *How did you arrive at the intraclass correlation coefficient?*

We provide some additional detail here, also referring the reader to the published open access protocol for our complete methodology.

As described in detail in the published protocol,¹ we adopted a conservative intraclass correlation coefficient (ICC) of 0.06, based on the ICCs for length of stay and appropriateness of stay in trials of acute care pathways.⁹

9. *Provide extra clarity on the sample size and the data extraction services involved.*

We have added the explicit detail the reviewer needed at these points.

With a two-sided $p < 0.05$, a sample size of 7840 patients was needed to detect a 2-14% absolute risk reduction...

Hospital peer group data were generated by the data extraction services *CHKs* (Alcester, UK) and *Dr Foster* (London, UK), also blinded to intervention group.

10. *What adjustments were made in the model?*

We list in the Methods section the numerous factors that were used as covariates in the model. Rather than duplicating this entirely in the Results (which would add to the length of the paper), we remind the reader again that the model incorporated both patient-level and ward-level characteristics. We would be happy to amend this further if the Editor feels it necessary.

Patient-level outcomes were adjusted for time effects, age, Charlson comorbidity index,¹⁰ and palliation status, as well as ward admissions. Ward-aggregate outcomes were adjusted for time effects, seasonal trends, the median Charlson score of patients on the ward that month, and the rotation of interns between departments. Survey analysis defined HEADS-UP participation as self-reported engagement in five or more briefings, and adjusted for self-reported workload on the NASA Task Load Index scale,¹¹ as well as hospital site and time period.

When adjusted for patient- and ward-level covariates, the intention-to-treat model showed increasing eLOS with PCTS...

11. *What does this current research add to the existing systematic review?*

We dedicate part of the Discussion to how this study contributes to the broader literature. We note that the effects of this type of interdisciplinary intervention are disputed. The varying impact of these interventions may be due, at least in part, to the variable implementation fidelity we describe here.

Earlier studies of prospective clinical surveillance did not evaluate an effect on patient outcomes, and so we can only position our work within a broader literature on ward-level interdisciplinary interventions. The variable implementation of these interventions likely contributes to their disputed impact on length of stay.^{12,13} In addition, the experience of PCTS implementation mirrors that of other structured team initiatives. Only 62% of mandated interdisciplinary surgical checklists are completed in their entirety.¹⁴ Similarly, structured bedside rounds may be implemented for only 54% of patients, despite co-creation with frontline staff.¹⁵ The benefits of these interventions are only identified at high fidelity,¹⁴ and so optimising their implementation should remain a focus for further research.

Reviewer 2

1. *This study is impressive throughout; classic stepped wedge trial study design.*

We were very pleased to read this reviewer's encouragement. We have tried to harness the strengths of the original manuscript to the additional detail requested by the other reviewers.

Reviewer 3

1. *The authors have failed to adjust for time in their analysis.*

The model did incorporate an adjustment for time, described in the original manuscript as examining 'temporal trends'. This may not have been clear enough, and we now use the phrase 'time effects'. We fully agree that this adjustment is critical for a stepped wedge study, and we now report the time effect specifically in the Results section.

There was an underlying trend of improved performance as the study period progressed (OR 0.98, 95% CI 0.96-0.996, p=0.016).

2. *The non-randomised nature of the study should be made clearer.*

We have updated the Abstract and the manuscript to emphasise this further

Prospective, stepped wedge, non-randomised, cluster controlled trial... [Abstract]

This was a non-randomised study, adopting a necessarily pragmatic approach to the order in which participating teams introduced the intervention. [Strengths and limitations]

We sought input from senior ward staff and nursing leadership, trying to identify when they felt they could support the intervention's introduction. It would have been counter-productive to insist on a fully randomised implementation schedule – although in practice, personnel and organisational changes meant that the anticipated leadership support varied. [Methods]

3. *The eligibility criteria for the clusters should be clear.*

We have addressed this in an amendment to the original manuscript.

Medical ward teams with an existing structure for daily interdisciplinary team meetings, and their managers, were invited to take part. All the approached teams agreed to participate.

4. *Confidence intervals are quite wide and it does not necessarily follow that the trial is well powered.*

We have removed the statement describing the study as well powered, as per the reviewer's request.

5. *The background should include justification for use of a stepped wedge design.*

The manuscript section on study design has been expanded to justify the stepped wedge approach. The manuscript is linked again to the published (open access) protocol, where further details are reported.

The trial was described in a published protocol.¹ Interdisciplinary ward teams were assigned to a multifaceted quality improvement intervention. Wards introduced the intervention at staged intervals over the study period, such that (by the end of the trial) all teams had adopted the intervention and contributed both control- and intervention-group data. Stepped wedge designs are increasingly used to evaluate service-level interventions in acute care.¹⁶⁻¹⁸ The stepwise implementation protocol is helpful when simultaneous rollout of the intervention would be impractical for logistical reasons.¹⁹

6. *A description of setting / control condition is missing.*

We now give a fuller description of the setting, in addition the summarised characteristics in Table 1.

Characteristics of the two participating institutions are given in Table 1. Most study wards were in a community general hospital in London. Both institutions faced significant challenges during the study period, with significant turnover in senior staff, mounting service pressures and financial restrictions. One hospital was in the process of a merger; the other was introducing an electronic health record. Neither managers nor clinical staff had protected time for quality improvement initiatives. Participating teams were teaching teams with attendings (consultants), residents (specialty trainees), and interns (foundation doctors). Interdisciplinary ward teams also included ward clerks and nurses, who were typically based on a single ward. Doctors, physiotherapists, and occupational therapists were based largely on one ward, but with some patient commitments elsewhere in the hospital.

7. *A schematic representation of the study is needed.*

This is provided in the supplement.

A schematic of the ward-level implementation schedule is provided in the supplement [supplement Figure 1].

8. *The given reference does not link clearly to a sample size methodology.*

The relevant section now gives the specific link to the paper describing our methodology.

Power calculations were conducted for the primary outcome (eLOS), using a recommended methodology for stepped wedge trials.²⁰

9. *Intra-class correlation coefficient needs to be reported for the primary outcome.*

We now state this in the Results.

The ICC for eLOS was 0.07 (95% CI 0.018-0.250).

10. *The paper needs absolute measures of treatment effects.*

The absolute treatment effect is now given in the Abstract and Results.

1714/6518 (26.3%) intervention admissions had eLOS vs 1279/4927 (26.0%) control admissions, an absolute risk increase of 0.3%.

11. Were all sub-groups and secondary outcomes pre-specified?

All the outcomes were pre-specified, as was the analysis of high-fidelity PCTS implementation.

Secondary outcomes (and the per protocol analysis described below) were pre-specified.

Reviewer 4:

1. *The conclusions do not focus on the primary outcome; the main conclusion should be that PCTS increased eLOS. The discussion can go on to explore the reasons for this.*

The Discussion now focuses much more on the intention-to-treat outcome, and why the disparity between the intention-to-treat and per protocol analyses may have come about.

In this stepped wedge cluster controlled trial, intention-to-treat and per protocol analyses produced conflicting results. eLOS increased overall, but was reduced by high fidelity PCTS implementation. Amongst the secondary outcomes, safety and teamwork climates improved with sustained PCTS implementation. High fidelity PCTS increased both incident reporting by non-nursing staff, and the number of non-falls reports.

There are several possible explanations for the tension between the two analyses. First, faithful attention to quality improvement efforts may indeed result in worse outcomes, perhaps by distracting attention from existing good practice.²¹ This is unlikely to have been the mechanism here: the wards that dedicated most time and effort to the intervention saw improved outcomes. Second, there may have been unmeasured confounders distinguishing between high fidelity and low fidelity PCTS teams. Teams that participate more wholeheartedly in trials may be more innovative, with strong leadership, a readiness for change, and better managerial relations.²² It is difficult, therefore, to entirely separate the intervention's effects from the characteristics of the teams that implemented it best. However, the study design mitigated for this, allowing for varying implementation fidelity by a single team across the study period. As the design elicited high fidelity ward *months* rather than high fidelity teams, unmeasured team characteristics should therefore hold less influence over the results. It is unlikely that our results are a simple reflection of teams' pre-existing practice or fluctuations in their workload.

The most likely explanation is that the intervention was used differently by high fidelity and low fidelity wards. In the linked qualitative analysis, high fidelity PCTS improved team autonomy in resolving problems, and facilitated managerial resolution where necessary.² By contrast, low fidelity teams may have used PCTS simply to record issues with a degree of frustration, without any positive change in their attitudes or behaviours.² The differential uptake of improvement interventions by teams that are already performing well is a difficult challenge to resolve.³ Rather than enacting change for themselves, low fidelity teams perhaps deferred to facilitation and managerial input, believing they would be sufficient to address unit-level problems. Previous quality improvement programmes have also inadvertently blurred lines of responsibility.⁴² Whilst imperfect PCTS implementation might be deleterious, our results suggest that interdisciplinary quality and safety interventions – when implemented well – remain a viable route to improvement.

2. *Please clarify if this is a per protocol analysis, in which case remove moderate fidelity.*

As requested, moderate fidelity is no longer specified in the paper; we limit the per protocol analysis purely to high fidelity implementation. There is no substantial change in the results.

3. *Please add more detail for how the survey data were collected and analysed. Did you do a difference-in-differences analysis? How did you justify ignoring clustering?*

The section on data collection now provides this detail. As described in the manuscript, the survey analysis incorporated a site-specific interaction effect between the intervention and time, i.e. a difference-in-difference analysis. We now include a sensitivity analysis adjusting for ward-level clustering, in which this interaction effect remains significant.

Potential participants in the HEADS-UP briefings (doctors and senior ward nurses) were invited to complete anonymous surveys at baseline and six months later, either by submitting responses electronically (via Survey Monkey) or via a paper questionnaire [Supplement]. Overall response rates were calculated using contemporaneous staffing rosters.

The significant improvement persisted in a sensitivity analysis that adjusted survey results for ward-level as well as hospital-level clustering.

4. *The range of implementation fidelity values may not be informative; an interquartile range would give a better feel for the spread of the data.*

We have changed the manuscript to quote the median and interquartile ranges for briefing frequency and engagement.

There was variation in implementation fidelity, both in terms of briefing frequency (median 78.9% working days/month, interquartile range 30.7%), and engagement (median 70 issues/ward/month, interquartile range 78.5).

5. *More detail is needed for Figure 1.*

As we explain in the manuscript, Figure 1 only provides an exploratory plot to help make sense of the data. As the reviewer notes, this is a modelled result, and we now provide more detail about how the model was generated. We found this Figure particularly helpful in view of the conflicting intention to treat and per protocol results. Our definition of implementation fidelity incorporated briefing frequency and engagement, and the Figure demonstrates how the two interacted to affect the primary outcome.

An exploratory plot of the modelled relationship between engagement, briefing frequency and the probability of eLOS highlighted different outcomes with high fidelity implementation and lower fidelity implementation [Figure 2]. The plot models engagement and briefing frequency as continuous variables within a multi-level model to help visualise their interaction.

6. *What is meant by 'mixed methods evaluation' in the Abstract?*

The reviewer rightly points out that this article does not include a qualitative component. A qualitative study is described in a linked article in this Journal; we have therefore removed the 'mixed methods' description here.

VERSION 2 – REVIEW

REVIEWER	Judith Poldervaart University Medical Center Utrecht, The Netherlands
REVIEW RETURNED	03-Mar-2017

GENERAL COMMENTS	All previous comments are adequately addressed by the authors, I have no further comments.
--

REVIEWER	Karla Hemming Uni of Birmingham UK
REVIEW RETURNED	16-Feb-2017

GENERAL COMMENTS	The paper is much improved and I have only minor comments:  1. The title should include the word non-randomised, so it is clear at the very outset that this is a report of an observational study. 2. Higher ICCs are not always conservative in a SW study – a higher ICC can mean a loss of power. See Girling and Hemming Stats in Med 2016 3. It is unclear what the difference in difference method is. I think the authors are referring to evaluation of the difference in some score between pre and post measurement times – this is well known to be a poor analysis method, and ANCOVA methods (i.e. adjustment for pre measurements as a covariate) is much more preferable. See Frison and Pocock circa 1997 4. How were time effects allowed for – using categorical effects for each measurement period? 5. Line 21 on pp14 I think the authors are conflating no effect with no evidence of an effect.
---

REVIEWER	Jennifer Thompson London School of Hygiene and Tropical Medicine, UK
REVIEW RETURNED	16-Feb-2017

GENERAL COMMENTS	My original review was not responded to in full. As a result, many of my original concerns remain. Please remove results from unadjusted analyses. These were not specified in the protocol and, as you state in the paper, there are strong time trends in your data that will confound the intervention effect. Were time trends included as linear effects? It is standard in SWTs
--

to count time effects as categorical or explore non-linear time effects, for example with fractional polynomials. I would not expect time trends to be linear in a hospital setting over such a long period. Please add a sensitivity analysis with one of these more standard analysis approaches.

Some of the secondary outcomes are only reported as per-protocol results in the text (although intention to treat results are given in table 3). Please add the intention to treat results for incident reports and the survey responses to the text.

Please add some more detail to the study design section. Although, you do state that there were 7 wards, you do not state how many steps the SWT had. How many months were there between each ward getting the intervention? How much follow up was there before any wards had the intervention and how long once all wards had the intervention?

Please add a limitation that the survey was a before and after analysis and so were confounded with time trends.

Page 13 final paragraph, “data were available for 71/73...” why were there 73 ward months? It is not clear how long each ward spent in control and then intervention, so it is not clear how you end up with 73 ward months.

My previous comment asking for the interquartile range was unclear. I meant it would be more informative to report the 25th and 75th percentiles of the data.

What is the difference between time effects and season trends that the ward-aggregate outcomes are adjusted for?

What do you mean by “sustained implementation” (page 15 paragraph 2)?

Figure 1 is still unclear. The effects are clearly modelled as non-linear, how was this done? Why isn't there a moderate fidelity line?

I disagree with conclusion that the results are unlikely to reflect fluctuations in workload. This seems like a possible explanation for your results. In ward/months with lower work load the HEADS-UP intervention was done with high fidelity and was beneficial to resolving issues in the wards. In ward/months with high work load the intervention was performed more detracted the limited time and resources from patient care and so had a negative impact on the ward. The explanation given in paragraph 2 of page 16 would make sense if the intervention had no impact overall, but the overall impact was negative. Please comment.

VERSION 2 – AUTHOR RESPONSE

Reviewer 1

1. *All previous comments are adequately addressed by the authors.*

We thank this reviewer for her time and contribution and are pleased she found our previous revisions acceptable.

Reviewer 3

1. *The title should include the word non-randomised.*

We have included this in the manuscript title.

Translating staff experience into organisational improvement: the HEADS-UP stepped wedge, cluster controlled, non-randomised trial.

2. *Higher ICCs are not always conservative in a stepped wedge study – a higher ICC can mean a loss of power.*

We agree that higher ICCs can reduce the study power, and it was our intention to highlight this. We have removed the word 'conservative' to avoid confusion.

We adopted an intraclass correlation coefficient (ICC) of 0.06, based on the ICCs for length of stay and appropriateness of stay in trials of acute care pathways.

3. *The difference-in-difference method is unclear.*

We now reference an explanatory article on the difference-in-difference method. As we explain, this is not an uncontrolled pre-post comparison. Non-participants were used as a control group at each survey point. This supports the argument (and statistical analysis) that changes in survey responses over time were due to the intervention, rather than any confounding background trend.

General linear regression models were used to analyse survey data, using a difference-in-differences approach (Dimick and Ryan, 2014) with an interaction code for time*intervention at each site (SPSS, v22). This method evaluated whether changes in survey responses over time differed between PCTS participants and non-participants.

4. *How were time effects allowed for – using categorical effects for each measurement period?*

Both Reviewers 3 and 4 raise this question. Although our primary analysis utilised time as a continuous variable for each measurement period, we have conducted the sensitivity analysis suggested by Reviewer 4. Classifying the measurement periods as a categorical variable, the results were no different. The direction of the effects, and the effect sizes, were unchanged – and remained statistically significant. This was the case both for the intention-to-treat and per protocol analyses.

A sensitivity analysis (coding time effects as a categorical variable) confirmed the effect sizes of both intention-to-treat and per protocol analyses, and their statistical significance.

5. *The authors are conflating no effect with no evidence of an effect.*

We have amended the sentence in question for clarity.

In the unadjusted model, eLOS did not change significantly with PCTS (OR 1.07, 95% CI 0.96-1.19, p=0.219).

Reviewer 4:

12. *Remove results from unadjusted analyses.*

We feel it is standard practice to explore the evolution of the results in unadjusted and adjusted analyses. The revised manuscript makes it clear that the adjusted analysis is definitive. Indeed, the unadjusted analysis does not even feature in the Abstract. We take the view that providing the unadjusted and adjusted data in the full Results section appropriately helps the reader understand the actual impact of adjustment.

1714/6518 (26.3%) intervention admissions had eLOS vs 1279/4927 (26.0%) control admissions, an absolute risk increase of 0.3%. PCTS increased eLOS in the adjusted intention-to-treat model (OR 1.32, 95% CI 1.10-1.58, p=0.003). [Abstract]

13. *Some secondary outcomes are [only] reported as per protocol results in the text, although intention-to-treat results are [also] given in the Table.*

Together, the Table and text report all of the relevant outcomes. We have amended the text to explicitly include the intention-to-treat results, to further enhance clarity.

There was no change in incident reporting in the intention-to-treat analysis. By contrast, high fidelity implementation increased the total number of incident reports submitted each month...

14. *Please add more detail to the study design section... how many months were there between each ward getting the intervention... It is not clear how long each ward spent in control and then intervention.*

As per the previous response letter, a schematic of the ward-level implementation schedule is provided in the supplement. The figure provides all of these details for interested readers, without adding to the length of the paper.

15. *Please add a limitation that the survey was a before-and-after analysis and so was confounded by time trends.*

As explained above (please see Reviewer 3, point 3), we used a difference-in-difference analysis which accounted for time trends.

16. *It would be more informative to report the 25th and 75th percentiles of the data.*

We now include this information as recommended.

Implementation fidelity was variable, both in briefing frequency (median 80% working days/month, interquartile range 65-90%), and engagement (median 70 issues/ward/month, interquartile range 34-113).

17. *What is the difference between time effects and season trends?*

'Time effects' refers to the monthly measurement periods at which data were collected. 'Seasonal trends' refers to the four seasons of the year, which were coded as a categorical variable. Because wards experience distinct seasonal variations (most notably poorer outcomes in the winter months) we incorporated this into the ward-level outcomes model. Please note we conducted a further sensitivity analysis to ensure that seasonal trends had no impact on our patient-level (primary outcome) results.

18. *What do you mean by sustained implementation?*

The different levels of implementation at each participating hospital are described in the text, as before: it is not a technical term as such, but rather a qualitative description of the intensity and effort that went into the implementation effort at different study sites.

Successful implementation at hospital 1 was recognised by the Care Quality Commission (the UK's healthcare quality inspectorate), whose inspectors observed briefings in use and formally reported them as 'outstanding practice', their highest rating. More limited organisational attention at site 2 resulted in slower programme implementation. At that site, PCTS was only embedding into routine practice as the second survey period began.

19. *Figure [2] is unclear; the effects are modelled as non-linear. How was this done? Why isn't there a moderate fidelity line?*

As in the previous response letter, we explain the creation of figure in the main manuscript. We have also now amended the caption of the figure to detail the methodology of the lines of best fit, for further clarity.

The moderate fidelity line was removed (as were all references to a moderate fidelity group) in response to this reviewer's previous requests – we have simply followed your recommendation.

An exploratory plot of the modelled relationship between engagement, briefing frequency and the probability of eLOS highlighted different outcomes with high fidelity implementation and lower fidelity implementation [Figure 2]. The plot models engagement and briefing frequency as continuous variables within a multi-level model to help visualise their interaction.

Quadratic lines of best fit are shown; these had the highest coefficients of determination (R2).

20. *I disagree with the conclusion that the results are unlikely to reflect fluctuations in workload... the explanation given would make sense if the intervention had no impact over all, but the impact was negative.*

With respect, the interpretation that the Reviewer offers [that the intervention was performed with lower fidelity at times of high workload] does not match the observed reality of the study. In the initial version of the manuscript we wrote that '*Direct observation showed multiple teams using briefings meaningfully, even under periods of great pressure*'. We also now reference the linked qualitative evaluation of this study (accepted for publication in this Journal), which highlighted how differently low- and high-fidelity teams related to the intervention. These differences extended well beyond changes in workload. Workload was accounted for in the statistical model, and direct observation showed that workload alone could not explain how teams were using the intervention.

As before, we also offer an explanation for the potentially detrimental impact of the intervention, which is important to note. We note that clinical teams may have deferred to their managers or the PCTS facilitator, rather than addressing problems themselves as they would have done ordinarily.

In the linked qualitative analysis, high fidelity PCTS improved team autonomy in resolving problems, and facilitated managerial resolution where necessary.¹ By contrast, low fidelity teams may have used PCTS simply to record issues with a degree of frustration, without any positive change in their attitudes or behaviours.¹ The differential uptake of improvement interventions by teams that are already performing well is a difficult challenge to resolve.² Rather than enacting change for themselves, low fidelity teams perhaps deferred to facilitation and managerial input, believing they would be sufficient to address unit-level problems.

VERSION 3 – REVIEW

REVIEWER	Jennifer Thompson London School of Hygiene and Tropical Medicine, UK
REVIEW RETURNED	30-Mar-2017

GENERAL COMMENTS	My comments have now been adequately address and I recommend the article for publication.
---